# Recurrent HBV Integration Targets as Potential Drivers in Hepatocellular Carcinoma

**DOI:** 10.3390/cells10061294

**Published:** 2021-05-23

**Authors:** Selena Y. Lin, Adam Zhang, Jessica Lian, Jeremy Wang, Ting-Tsung Chang, Yih-Jyh Lin, Wei Song, Ying-Hsiu Su

**Affiliations:** 1JBS Science, Inc., Doylestown, PA 18902, USA; slin@jbs-science.com (S.Y.L.); jwang336699@gmail.com (J.W.); fsong@jbs-science.com (W.S.); 2The Baruch S. Blumberg Research Institute, Doylestown, PA 18902, USA; adamzhang0214@gmail.com (A.Z.); jessiellian@gmail.com (J.L.); 3Department of Internal Medicine, National Cheng Kung University Medical College, Tainan 704, Taiwan; ttchang@mail.ncku.edu.tw; 4Department of Surgery, National Cheng Kung University Medical College, Tainan 704, Taiwan; lyj007@mail.ncku.edu.tw

**Keywords:** hepatitis B virus, hepatocellular carcinoma, HBV integration, HCC driver identification

## Abstract

Chronic hepatitis B virus (HBV) infection is the major etiology of hepatocellular carcinoma (HCC), frequently with HBV integrating into the host genome. HBV integration, found in 85% of HBV-associated HCC (HBV–HCC) tissue samples, has been suggested to be oncogenic. Here, we investigated the potential of HBV–HCC driver identification via the characterization of recurrently targeted genes (RTGs). A total of 18,596 HBV integration sites from our in-house study and others were analyzed. RTGs were identified by applying three criteria: at least two HCC subjects, reported by at least two studies, and the number of reporting studies. A total of 396 RTGs were identified. Among the 28 most frequent RTGs, defined as affected in at least 10 HCC patients, 23 (82%) were associated with carcinogenesis and 5 (18%) had no known function. Available breakpoint positions from the three most frequent RTGs, *TERT*, *MLL4/KMT2B*, and *PLEKHG4B*, were analyzed. Mutual exclusivity of TERT promoter mutation and HBV integration into TERT was observed. We present an RTG consensus through comprehensive analysis to enable the potential identification and discovery of HCC drivers for drug development and disease management.

## 1. Introduction

Hepatocellular carcinoma (HCC) is the second leading cause of cancer deaths worldwide [1,2,3], and its poor prognosis is, in part, due to the lack of effective treatment options. A major etiology of this multifactorial disease is chronic hepatitis B virus (HBV) infection, which is associated with approximately 50% of HCC cases worldwide [4]. Current HBV–HCC screening guidelines vary regionally: the American Association for the Study of Liver Disease (AASLD), Canadian Association for the Study of the Liver (CASL), European Association for the Study of the Liver (EASL), Asian-Pacific Association for the Study of the Liver (APASL), and Latin-American Association for the Study of the Liver (ALEH) all have distinctive recommendations. Irrespective of the region, ultrasound with or without serum AFP screening is recommended every 6 months in HBV populations with risk factors that may include being of Asian descent, >40 years of age, cirrhosis, and PAGE-B score [5].

During infection, HBV can integrate into the host genome. It has been proposed that integration events mostly occur through non-homologous end joining (NHEJ) and micro-homologous recombination [6,7,8,9]. Although HBV DNA integration into the host genome is considered rare—with an estimate of one integration event per 10,000 HBV-infected hepatocytes [10]—integrated viral DNA has been reported in more than 85% of HBV-related HCC (HBV–HCC) cases, suggesting a significant association between HBV integration and hepatocarcinogenesis. Additionally, mechanisms of HBV integration in HCC carcinogenesis potentially vary from patient to patient, including the insertional mutagenesis of HCC-associated genes, induction of chromosomal instability, and continuous expression of viral proteins [11,12]. Therefore, understanding the impact of integrated HBV DNA on carcinogenesis may identify potential HCC driver genes as personalized biomarkers could paving the way for precision disease management in HBV–HCC patients.

With the advent of next-generation sequencing (NGS), thousands of HBV integration sites have been identified across the human genome. Although no host sequence preference or specificity [13,14,15,16,17,18] has been identified, integration can activate known HCC driver genes and has been reported in *TERT, CCNE1,* and *MLL4* [19]. These, and other frequently affected genes, have become known as recurrently targeted genes (RTGs). Similar to the approach of identifying *BRAF* V600E driver mutations as recurrent hotspot mutations, here, we take advantage of the large number of reported integration sites from the literature and our in-house study to test the hypothesis that characterization of frequent RTGs can be a tool for the identification of HBV–HCC drivers.

In this study, we collected integration sites identified in HCC tumor tissue and defined RTGs. A total of 18,596 HBV integration sites, detected using PCR- and NGS-based approaches [5,12,13,14,15,16,17,18,19,20,21,22,23,24,25,26,27,28,29,30,31,32,33,34,35], were analyzed. By characterizing the most frequent RTGs, we demonstrate the potential of identifying HCC drivers for HCC precision medicine and drug development.

## 2. Materials and Methods

### 2.1. Data Mining/Search Strategy

We searched the PubMed (1 January 2000–1 December 2020) database using Medical Subject Heading (MeSH) terms “hepatitis B virus”, “HBV integration”, and “hepatitis B integration sites” to identify studies that have reported HBV integration sites by either NGS- or PCR-based approaches. Additional studies were obtained by cross-referencing from the literature. We included only English-language studies with HCC subjects. We included all such studies that identified HBV integration sites using NGS-based approaches. Among studies using PCR-based methods, we only included those with sample sizes of 10 or more HCC patients. HBV integration sites identified by RNA-seq [7,8,20] were not included, because the expression of integrated sequences can be regulated by many host cellular factors and is not within the scope of this study. We filtered out repeated integration sites to ensure that each integration site was included only once, except for two studies that utilized different methods on overlapping samples [21,22]. A total of 33 reported studies in addition to our study are included, as summarized in Appendix A.

### 2.2. In-House HCC Specimens and HBV Integration Analysis

Archived FFPE tumor tissue DNA (Appendix A) [23,24] from stage I–IIIB patients (*n* = 32) was obtained from the National Cheng-Kung University Medical Center, Taiwan, and collected following the guidelines of the Institutional Review Board. An HBV enrichment NGS assay (JBS Science, Doylestown, PA, USA) was used to capture integration sites. Briefly, NGS libraries were generated and enriched for HBV sequences, and the enriched libraries were sequenced on the Illumina MiSeq platform (Penn State Hershey Genomics Sciences Facility at Penn State College of Medicine, Hershey, PA, USA). The NGS data were analyzed using ChimericSeq [25] to identify HBV–host junction sequences. Reference genomes NC_003977.1 (HBV) and GRCh38.p2 (human) were used. The closest genes within 150 kb of the HBV integration breakpoint were identified with ChimericSeq software. Genes were defined as in NCBI’s RefSeq database. Tailored junction-specific PCR-Sanger sequencing assays were designed to validate each HBV integration site of interest, identified by the HBV enrichment NGS assay.

### 2.3. TERT Promoter Mutation Analysis by PCR-Sanger Sequencing

HCC tissue DNA was used to amplify a 163 bp region (Chr5:1,295,151–1,295,313) of the TERT promoter by HotStart Plus Taq Polymerase (Qiagen, Valencia, CA, USA) with forward primer 5′-CAGCGCTGCCTGAAACTC-3′ and reverse primer 5′-GTCCTGCCCCTTCACCTT-3′. The PCR products were sequenced at the NAPCore Facility at the Children’s Hospital of Philadelphia (Philadelphia, PA, USA) and analyzed using the ClustalW software [26].

### 2.4. Identification of RTGs

To identify host genes that may have been affected by HBV DNA integration across all studies, we identified the closest gene within 150 kb of the integration event, the reported distance within which host genes can be impacted by integration [27,28]. For a gene to be classified as an RTG, it must have been identified as an HBV integration locus in two or more independent studies from at least two independent laboratories, each of which included at least two HCC patients with the gene affected in the tumor tissue. The requirement that a gene must have been identified in multiple laboratories served to minimize the issue of potential cross-contamination. The full list of identified RTGs can be provided upon request.

### 2.5. Gene Functional Enrichment Pathway Analysis

A total of 396 RTGs were subjected to enrichment pathway analysis using Enrichr (http://amp.pharm.mssm.edu/Enrichr (accessed on 30 January 2021)), to identify significantly (*p* < 0.05) enriched pathways as determined by gene ontology.

## 3. Results

### 3.1. Overview of the Studies for RTG Identification

The studies included in RTG identification are summarized in Table 1. Twenty-three studies utilized NGS, including our in-house study (Appendix A), and 10 studies utilized PCR-based approaches for HBV integration identification. For each study, the sample size and the number and percentage of HCC tumors that had detectable integration sites are listed. We compiled a total of 18,596 HBV integration sites from tumor tissues of a combined 1310 HCC patients. In all, we found that 76% of tumor tissues (*n* = 1733) contained detectable integration sites. In the 12 studies that enriched for the whole HBV genome, on average, 72% (range 54–100%) of the tumors examined were found to have integrated HBV DNA (*n* = 880). In two studies using an HBV DR1-2 enrichment NGS assay, 65% [29] and 91% (our study) of tumors examined were positive for integrated HBV DNA.

### 3.2. Characteristics of HBV–HCC Patients with Integrated HBV DNA

To compare the general sociodemographic and clinical characteristics of the HBV–HCC population compiled in this study (age, gender, HBV genotype, and cirrhotic etiology (designated as “cirrhotic HCC”)) to those previously reported in the literature [4,58,59], we divided our study population into two groups based on the detectability of integrated HBV DNA (Table 2). Analysis of each parameter was performed where available. Overall, there were no significant differences between the two groups (with integrated DNA detectable vs. not detectable), including in the sex ratio. Of the three reported HBV genotypes, genotype C was the most frequently reported in the integration-positive patient cohort (75%), while the cohort with no detectable integration had only two patients with a genotype reported (both C). Interestingly, only 47% (7/15) of the integration-negative patients had cirrhosis, which is much lower than the 70–90% range previously reported in the general HBV–HCC population [4]. In the integration-positive cohort, 62% of HCC tumors were derived from a cirrhotic liver, which is also below the previously reported range.

Lastly, because it is known that HBV behavior differs in Asian and Western cohorts [60], the geographic region where patient specimens were collected is indicated for each study. The majority of studies (72.7%) collected patient specimens in Asian populations, while only 6.0% examined Western cohorts.

### 3.3. Recurrent Sites of HBV Integration

Next, we identified RTGs in the compiled HCC cohort using the criteria described in the Materials and Methods and then examined their associations with carcinogenesis from the published literature. Of the 18,596 integration sites examined, 2892 sites were found within 150 kb of gene-coding sequences, and 396 were RTGs. The 396 RTGs were found in 673 HCC patients, which was 51% of the integration-positive patients (*n* = 1310) or 39% of all HCC patients (*n* = 1733). To investigate the association of frequent RTGs with carcinogenesis, we analyzed the most frequently recurrent genes (n = 28), defined as found to contain HBV integrants in at least 10 HCC patients (Table 3). As expected, TERT and MLL4 were the two most recurrent genes. Interestingly, these 28 genes either have previously been associated with carcinogenesis (23/28, 82%), or have no known function (5/28, 18%). A full list of biological pathways associated with the 28 RTGs is detailed in Appendix A.

Next, the 396 RTGs were queried for significantly enriched gene ontology (GO) pathways using Enrichr [61]. The top enriched biological pathways were axon guidance (*p* = 0.00007) and positive regulation of Ras protein signal transduction (*p* = 0.0001), suggesting possible links with oncogenesis (Figure 1A). Heparin sulfate-glucosamine 3-sulfotransferase I (*HS3ST1*) activity was among the top enriched pathways from GO molecular functions (Figure 1B). Sulfotransferases have reported associations with carcinogenic activity, and *HS3ST1* in particular has been implicated in inflammation [62]. Finally, a search of the Drug Signatures Database (DSigbDB) identified trichostatin, which selectively inhibits class I and II histone deacetylases (HDACs), as the drug/compound related to most RTGs (115/396; Figure 1C).

**Table 3 cells-10-01294-t003:** Most frequently reported genes with HBV integrants in HCC tumors.

RTG	Full Gene Name	Subjects (*n*)	Junctions (*n*)	Cancer Associated [Ref]
*TERT* ^1^	Telomerase reverse transcriptase	357	561	Multiple cancers [63]
*MLL4* (*KMT2B*)	Lysine methyltransferase 2B	130	220	HCC [52,64], Spindle cell sarcoma [65], Gastric cancer [66]
*PLEKHG4B*	Pleckstrin Homology and RhoGEF Domain Containing G4B	38	115	Neuroblastoma [67]
LOC100288778	WAS protein family homolog 8 pseudogene	34	79	SCLC [68]
*DDX11L1*	DEAD/H-Box Helicase 11 Like 1	33	57	Function unknown
*CCNE1*	Cyclin E1	31	54	Multiple cancers [69]
*CCNA2*	Cyclin A2	26	45	Multiple cancers [70]
*SNTG1*	Syntrophin Gamma 1	25	27	Lung adenocarcinoma [71]
*PGBD2*	PiggyBac Transposable Element Derived 2	22	51	Function unknown
*DUX4L4*	Double homeobox 4 like 4 pseudogene	20	35	DUX4 Ewing’s sarcoma [72], ALL [73]
*ROCK1P1*	Rho Associated Coiled-Coil Containing Protein Kinase 1 Pseudogene 1	20	35	Prostate cancer [74]
*ANKRD26P1*	ankyrin repeat domain 26 pseudogene 1	19	72	Breast cancer [75]
*PARD6G*	Par-6 Family Cell Polarity Regulator Gamma	19	42	Breast, kidney, liver, lung, ovary, and pancreatic cancers [76]
*FN1*	Fibronectin 1	17	18	Multiple cancers [77]
*CWH43*	Cell Wall Biogenesis 43 C-Terminal Homolog	14	73	CRC and TSHomas [78]
*TPTE*	Transmembrane Phosphatase with Tensin Homology	13	30	HCC [79], prostate cancer [80]
*FAM157A*	Family with Sequence Similarity 157 Member A	14	22	Function unknown
LOC728323/LINC01881	Long Intergenic Non-Protein Coding RNA 1881	13	22	Oral cancer [81]
*EMBP1*	Embigin pseudogene 1	12	27	Oropharyngeal carcinoma [82], multiple primary cancers [83]
*OR4C6*	Olfactory Receptor Family 4 Subfamily C Member 6	12	22	Pancreatic cancer [84]
*PRMT2*	Protein Arginine Methyltransferase 2	12	15	Glioblastoma [85]
*ROCK1*	Rho Associated Coiled-Coil Containing Protein Kinase 1	12	23	HCC [86,87,88,89], CRC [90]
*ANHX*	Anomalous Homeobox	11	16	Function unknown
*CTNND2*	Catenin Delta 2	11	14	HCC [91,92], prostate cancer [93], lung cancer [94]
*DDX11L9*	DEAD/H-Box Helicase 11 Like 9 (Pseudogene)	11	16	Function unknown
*SENP5*	SUMO Specific Peptidase 5	11	11	HCC [95], breast cancer [96]
*ZNF595*	Zinc Finger Protein 595	11	14	Lung cancer [97], Gastric cancer [98]
*CDRT7*	CMT1A Duplicated Region Transcript 7	10	10	Glioma [99]

^1^ The number of reported *TERT* junctions is slightly skewed, because one of the references [45] only reported the number of *TERT* promoter integrations. RTG, recurrently targeted gene; HCC, hepatocellular carcinoma; NSCLC, non-small cell lung cancer; SCLC, small cell lung cancer; ALL, acute lymphocytic leukemia; CRC, colorectal cancer; TSHoma, thyrotropin-secreting pituitary adenoma; CLL, chronic lymphocytic leukemia; RCC, renal cell carcinoma.

### 3.4. Integration Breakpoints in the HBV Genome

To investigate the distribution pattern of the integration breakpoints in the HBV genome, we analyzed HBV breakpoints in tumors (*n* = 4008) where available. We omitted studies that enriched for HBV DR1-2 sequences to assess the HBV breakpoint distribution in an unbiased manner. Consistent with previous reports [6,39,43], we observed that 39.2% of all breakpoints were within the HBV nt. 1300–1900 region in tumors. This region is at the 3′ end of the *HBx* gene and contains the initiation site of viral replication/transcription. Additionally, we observed a breakpoint hotspot in the HBV DR1-2 region, representing 29.6% of all HBV breakpoints (Figure 2).

### 3.5. Genomic Breakpoints of TERT, MLL4 and PLEKHG4B RTGs

HBV integration is believed to be non-sequence-specific; therefore, it was of interest to examine all RTG breakpoint coordinates for similarity to each other. To do so, we plotted the available human and HBV breakpoint coordinates of the three most frequent RTGs identified, *TERT*, *MLL4*, and *PLEKHG4B* (Figure 3).

For *TERT*, the most frequent RTG, 291 of 561 junctions from 205 HCC patients, had both human and HBV breakpoint coordinates available. As expected, most of these breakpoints were centered between DR2 and DR1 of the viral genome and were highly concentrated at the promoter region of the *TERT* gene (Figure 3A). Of note, 142 (48.7%) *TERT* integration junction sequences were unique, i.e., reported from 142 different subjects, whereas the rest (57 sequences) were each found in two or more HCC patients. Of the 477 available breakpoint coordinates in the *TERT* gene, 361 (75.6%) junctions were located upstream of exon 1 and, of these upstream breakpoints, 242 (50.7%) were located within the *TERT* promoter region (Chr5:1,295,162–1,296,162).

*MLL4* was the second most frequently reported RTG, with 220 junctions identified from 130 HCC patients. Among them, 150 breakpoints from 83 HCC patients had both human and viral coordinates available and are plotted in Figure 3B. As with *TERT*, most of the breakpoints were clustered between DR2 and DR1 of the viral genome and concentrated within exon 3 of the *MLL4* gene. Eleven different recurring breakpoints were found in two or more HCC patients, accounting for 37 of the 150 junctions examined.

The third most reported RTG was PLEKHG4B, with 116 integrations reported in this study. Interestingly, all the 116 breakpoints were centered within a 3 kb region which was 131 kb upstream from the *PLEKHG4B* coding region. A total of 47 breakpoints from eight HCC patients had both viral and human coordinates available, as shown in Figure 3C. All the breakpoints were found upstream of the transcription start site (Chr5:140,373). Unlike in *TERT* and *MLL4*, the viral breakpoints were centered in two HBV regions (nt. 1802–1814 and 2390; 15 and 14 breakpoints, respectively) but at various human coordinates. Further analysis of the human sequences (Chr5:10,000–13,000) in the upstream integration region revealed a 1877 bp segment containing simple repeat sequences and a 1057 bp segment containing satellite sequences. This finding is interesting because HBV has been suggested to have a higher propensity to integrate into repeat regions/retrotransposons, as recently shown by Chauhan et al. [100]. In addition, a motif, TAAACCCTAAC, was discovered, appearing four times in the Chr5:10,000–13,000 region and once in the HBV genome, each with *p* < 0.0001 by the Student’s *t*-test. However, querying the GenomeNet database for this motif produced no matches, and analysis of the region for known motifs produced no results. No recurrent breakpoints were identified among the integration sites in the *PLEKHG4B* gene. Notably, seven of the eight HCC patients with this unique junction pattern, where the same HBV breakpoint was joined to different host chromosome breakpoints, were reported from one study by Yang et al. [43].

*TERT* hotspot promoter mutations (−124, −146) are the most frequently reported mutations in HCC, found in about 50% of cases [101,102,103,104,105,106]. In HBV–HCC, up-regulation of TERT expression could also be caused by HBV integration at or near the *TERT* promoter region [28,29,30,32,38,48]. Therefore, we next compared the frequencies of *TERT* promoter mutation and HBV integration. For our in-house cohort (*n* = 22), shown in Figure 4A, promoter mutations were found in 6 of 22 samples, and integrations in the *TERT* gene were found in 5 of 22 samples in a mutually exclusive fashion. Together, *TERT* alterations were detected in 50% (11/22) of this cohort. To determine if this mutual exclusivity applied to a larger sample size, we examined *TERT* alterations identified by us and others [40,42,45] together, as summarized in Figure 4B. Of the 347 HBV–HCC patients, 174 (50%) were found to have detectable *TERT* alterations. Promoter mutations and HBV integrations were mutually exclusive and comprised 40% (70/174) and 60% (104/174) of all alterations, respectively.

## 4. Discussion

In this study, we compiled and studied a total of 18,596 HBV integration sites from 1733 HCC patients reported in 32 previous studies and our in-house study, to test our hypothesis that frequent HBV RTGs are associated with HCC carcinogenesis and thus may represent HCC driver genes that can reveal potential therapeutic targets and molecular profiling for precision medicine. Using three criteria for RTG selection, we identified 396 RTGs. Encouragingly, the most frequent RTGs (*n* = 28), reported in at least 10 HCC patients each, either have known involvement in carcinogenesis (23/28; 82%) or have no known function (5/28; 18%). By gene ontology analysis, RTGs were mapped to functions related to carcinogenesis. We describe a potential tool to identify or discover HCC drivers by the characterization of frequent RTGs. More studies are needed to demonstrate the association of carcinogenesis with the frequency of RTGs that have unknown functions.

Three criteria were applied to candidate genes to identify 396 HBV RTGs in this study: (1) location within 150 kb of the breakpoint, the reported distance within which host genes can be impacted by integration [27,28]; (2) the presence of HBV integrants in ≥2 HCC patients; and (3) identification in ≥2 independent laboratories to avoid the possibility of contamination within a laboratory. We are aware that the identification of RTGs across multiple studies is complex due to multi-faceted underlying variables such as integration detection methodologies and patient populations. For instance, some studies do not contain any of the 396 RTGs that we identified [54,55], while others report a high detection rate of a particular RTG, such as *MLL4* [52] or *cMYC* [39]. We are also aware that different methodologies for identifying integrations may have different sensitivities that can result in different integration site profiles. Furthermore, how DNA samples were prepared varied across the studies. Despite these limitations, the detection of highly frequent RTGs as potential HBV–HCC drivers could be clinically useful for HCC patients. Notably, our approach can only identify driver genes involved in HBV–HCC (rather than HCC in general) and is limited to insertional mutagenesis. In addition, HBV integration could contribute to carcinogenesis by causing chromosomal instability, expression of the oncogenic HBV protein X, and inflammation induced by other HBV proteins expressed from the integrated DNA. Although the most frequent RTG, *TERT*, is also the most frequently mutated gene in HCC in general, other frequently mutated HCC genes, such as *TP53* and *CTNNB1*, were not among the top 28 RTGs. These findings suggest that using RTG analysis as a tool to identify driver genes is limited to HBV–HCC.

By detailed analysis of integrations in the three most frequent RTGs (*TERT*, *MLL4*, and *PLEKHG4B*), we revealed three important features. First, as expected, most of the junction coordinates are different, supporting a non-sequence-specific mechanism of integration in the host genome. Among the junction sequences identified in multiple HCC patients, such as the 57 junctions described in this study, frequent integration in the promoter region of the *TERT* gene highlights the potential importance of the site in hepatocarcinogenesis. Secondly, an interesting pattern of integration into repetitive sequences was observed in the *PLEKH4G4B* junctions shown in Figure 3C. Furthermore, a highly repetitive sequence, satellite sequences, and the TAAACCCTAAC motif were identified in these regions, consistent with previous observations that integration occurred frequently in repetitive sequences [100]. However, because these unique integration sequence patterns were reported in only one study and validation in original tissue DNA was not reported, the possibility of an artifact has not been excluded. Lastly, the mutually exclusive detection of *TERT* promoter mutations and *TERT* integrations was shown in our small cohort of 22 HCC patients as well as in a larger compiled cohort of 347 HCC patients [40,42,45]. *TERT* promoter mutations account for 50% of all alterations in this gene, suggesting the importance of identifying *TERT* integration. This observation also further emphasizes the need for analysis of frequent RTGs to better characterize HCC.

In summary, we identified 396 RTGs from an analysis of 18,596 HBV integration sites. Twenty-eight of the RTGs were identified in 10 or more HCC patients. The RTGs were found to be significantly enriched in several GO pathways. The 28 most frequent RTGs have previously been implicated in carcinogenesis (23/28) or have no known function (5/28). Taken together, our findings demonstrate the potential of RTG identification as a tool for HBV–HCC driver discovery and tumor characterization, to be used in precision medicine and drug development. More studies are needed to further refine the criteria for RTG identification for its applications.

## Figures and Tables

**Figure 1 cells-10-01294-f001:**
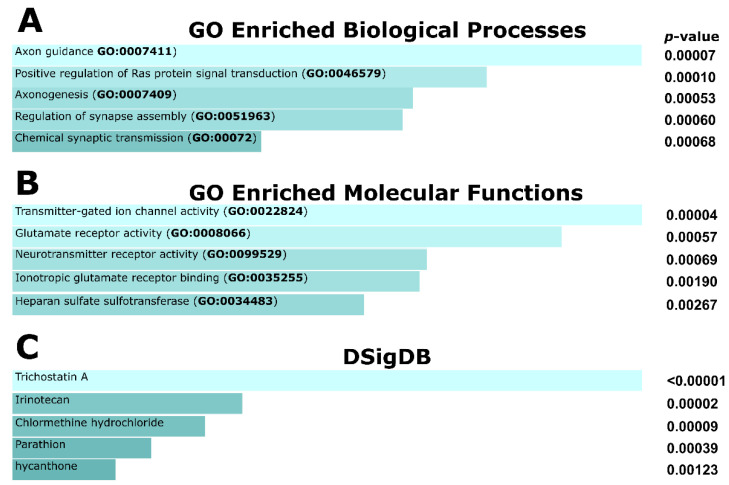
The top five significantly enriched gene ontology terms associated with RTGs as determined by the EnrichR software. (**A**) Biological processes, (**B**) molecular functions, and (**C**) Drug Signatures Database (DSigDB). Pathways are presented based on the combined EnrichR score. DSigDB relates drugs/compounds to their target genes.

**Figure 2 cells-10-01294-f002:**
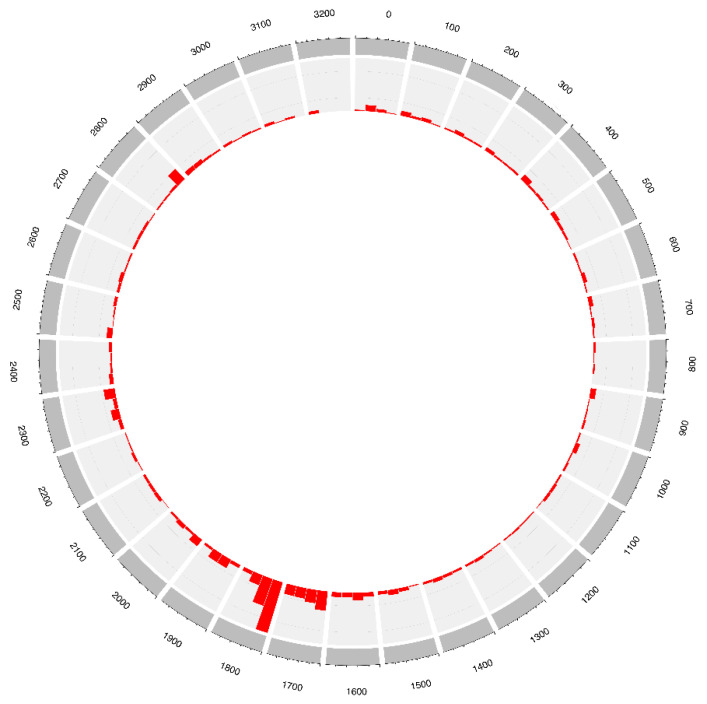
Distribution of integration breakpoints in the HBV genome in HCC tumor samples. A total of 3052 HBV breakpoints were plotted. The histogram represents the frequency of integration breakpoints at different loci in the HBV genome (nt. 1–3215; bin size 25) as numbered in the outer ring.

**Figure 3 cells-10-01294-f003:**
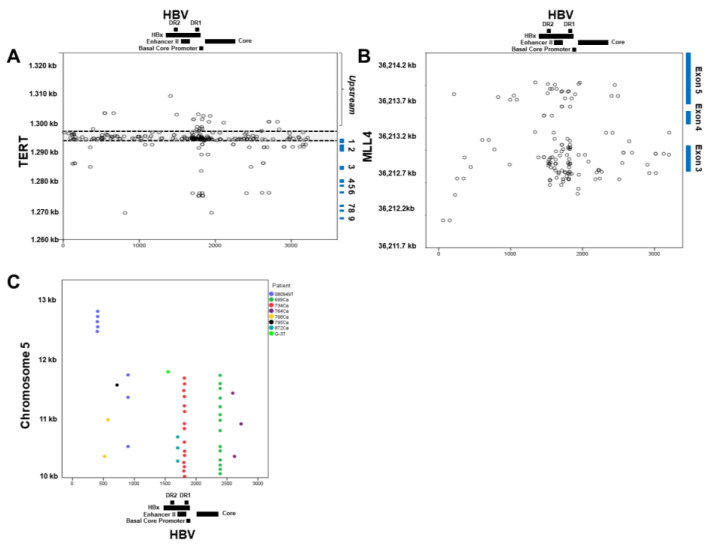
Mapping of *TERT, MLL4,* and *PLEKH4G4B* HBV integration breakpoints along the human and HBV genomes. (**A**) *TERT* breakpoints. A total of 219 *TERT* integration breakpoints derived from 161 patients are plotted. The *y*-axis coordinates decrease from 1320 kb to 1260 kb to represent the direction of the transcriptional start site from a 5′–3′ orientation. The expanded view of the region with the most integration sites is shown for the human genome position 1297 kb to 1294 kb and the HBV nt. 1500–2000. (**B**) *MLL4* breakpoints. A total of 115 *MLL4* integration breakpoints, derived from 64 patients, are plotted. Blue squares denoting exon regions are shown. (**C**) *PLEKH4G4B* breakpoints. A total of 47 of the 116 reported *PLEKHG4B* breakpoints plotted are derived from 8 unique HCC patients. Colored dots correspond to each unique patient. Each dot represents the mapped locations of the integration sites where the human gene breakpoints (GRCh37) are located on the *y*-axis, and HBV breakpoints are located on the *x*-axis, in accordance with the reported locations.

**Figure 4 cells-10-01294-f004:**
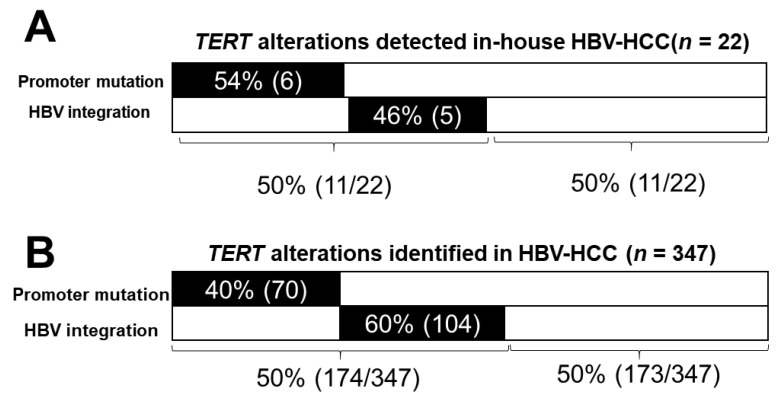
*TERT* gene alterations identified in HBV–HCC tissues. (**A**) In-house cohort (*n* = 22). (**B**) Compiled HBV–HCC cohort. Patients are derived from our in-house study (*n* = 22) and from the literature (*n* = 129) [40,42,45]. The number of HCC patients is indicated in parentheses.

**Table 1 cells-10-01294-t001:** Summary of HBV integration studies included in this analysis.

		Study	HCC Patients (n)	# of Subjects with Integrated DNA * (% of total)	# of Junctions * Identified in Subjects	Junction Sequence	Clinical Variables	Geographic Region
NGS-based	WGS	[21]	3	3 (100%)	15	Yes	Yes	NA ^7^
		[30,31]	91 ^1^	64 (45%)	223	Yes	Yes	Japan
		[32,33]	81	76 (94%)	344	Yes	Yes	Hong Kong
		[34]	2	2 (100%)	5	Yes	Yes	China
		[22]	5 ^1,2^	5 (100%)	92	Yes	-	NA ^7^
		[35]	5	5 (100%)	21	Yes	Yes	NA
		[36]	3	2 (67%)	11	Yes	Yes	China
	Whole HBV Genome	[37]	20	16 (80%)	424	Yes	Yes	China
		[38]	48	26 (54%)	57	-	-	Singapore
		[39]	60	51 (85%)	156	Yes	Yes	China
		[6]	426	344 (81%)	3486	Yes^3^	-	China
		[40]	49	28 (57%)	121	Yes	Yes	Japan
		[41]	40	35 (90%)	257	Yes	Yes	NA
		[42]	101	94 (93%)	510	Yes	Yes	Taiwan
		[43]	54	54 (100%)	2870	Yes	Yes	China
		[44]	1 ^5^	1 (100%)	2	Yes	Yes	NA
		[45]	95	85 (89%)	1563	Yes	Yes	Hong Kong
		[46]	243	72 (30%)	4347	Yes	Yes	Japan
		[47]	50	44 (88%)	289	Yes	Yes	Taiwan
	DR1-2	[29]	40	26 (65%)	42	Yes	Yes	China
		Our study	22	20 (91%)	27	Yes	Yes	Taiwan
PCR-based		[48]	13	2 (15%)	2	Yes	-	USA
		[49]	14	14 (100%)	14	Yes	Yes	China
		[50]	15	15 (100%)	15	-	Yes	NA
		[51]	60	55 (92%)	60	-	Yes	NA
		[52]	10	7 (70%)	8	Yes	Yes	Japan
		[53]	60	41 (68%)	101	Yes	Yes	China
		[54]	59 ^4^	45 (76%)	45	-	-	Italy
		[55]	15	9 (60%)	9	-	-	China
		[56]	18	18 (100%)	2083	-	Yes	China
		[57]	30	21 (70%)	1397 ^6^	-	-	China
		Total	1733	1310 (76%)	18,596			

^1^ HBV (+) HCC cohorts only. ^2^ Three patients previously reported in Jiang 2012 [21] were removed. ^3^ Only human chromosome sequence position provided. ^4^ Cohorts of HBsAg (−)/occult (+) and HBsAg (+) HCC patients. ^5^ HBV capture sequencing of a single occult HBV infection (OBI) patient. ^6^ Five of the thirty HCC tissues were analyzed by HBV-targeted NGS. ^7^ Specimens were purchased commercially from vendors such as Seracare LifeSciences, ProteoGenex, or Indivumed. * HBV DNA integration sites. WGS, whole-genome next-generation sequencing; Whole HBV genome, whole HBV genome enrichment performed prior to NGS; DR1-2, HBV DR1-2 integration hotspot region enriched prior to NGS; -, not available.

**Table 2 cells-10-01294-t002:** Major sociodemographic and clinical features of the study subjects with and without detectable integrated HBV DNA in tumor tissue.

	Reported HBV–HCC Population ^1^	Integrated HBV DNA	
		Not Detectable (*n* = 423)	Detectable (*n* = 1310)
Age (years) Range Avg. ± SD	NA 55–65 ± NA	33–83 59.4 ± 13.0 (*n* = 40)	11–85 54.9 ± 11.6 (*n* = 359)
Gender (Total)		(*n* = 66)	(*n* = 635)
Male	NA	40	466
Female		15	169
Genotype (Total)		(*n* = 2)	(*n* = 84)
B	NA	0	24 ^2^
C		2	76
D		0	1
Cirrhosis %	70–90% (*n* = NA)	46.7% (*n* = 7/15)	62.3% (*n* = 105/279)

**^1^** Characteristics of the HBV–HCC subjects obtained from previous reports [4,58,59]. **^2^** One patient contained a mix of HBV B and C genotypes. NA, not available; *n,* number of patients available for analysis; Avg. ± SD, average ± standard deviation.

## Data Availability

Detailed data are available upon request.

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
