# Peer review of "Recurrent HBV Integration Targets as Potential Drivers in Hepatocellular Carcinoma"

_cells, 2021, doi:10.3390/cells10061294_

Round 1

Reviewer 1 Report

The present manuscript should change the title,

Characterization of genes recurrently targeted by HBV integration in hepatocellular carcinoma: a potential tool for driver identification, the reader became bored with the verbosity.

The abstract should rewrite, could not list too many results in it.

Figure 1 and figure 2, figure 3 need regenerate. Should be clear and publishable.

Author Response

Point 1. Characterization of genes recurrently targeted by HBV integration in hepatocellular carcinoma: a potential tool for driver identification, the reader became bored with the verbosity.

Response: We have changed the manuscript title to Recurrent HBV integration targets as potential drivers in hepatocellular carcinoma”.

Point 2. The abstract should rewrite, could not list too many results in it.

Response: We have revised the abstract accordingly and marked the changes in red.

Point 3 Figure 1 and figure 2, figure 3 need regenerate. Should be clear and publishable.

Response: Figures 1-3 have been regenerated at 600 DPI.

Reviewer 2 Report

- I think impact of this manuscript could be enhanced by referring in the introduction to recent efforts focused on surveilling patients with HBV for HCC development (e.g. Anugwom CM, Allaire M, Akbar SMF, Sultan A, Bollipo S, Mattos AZ, Debes JD.  Hepatitis B-related hepatocellular carcinoma: surveillance strategy directed by immune-epidemiology. Hepatoma Res. 2021;7:23. doi: 10.20517/2394-5079.2021.06.)

-Table 3 would benefit from listing the full length name of genes, also some emphasis on what is not found would be interesting, like integrations in Axin1/2/Tankyrase genes or so. Also the cancer associated annotations feel a little crude as for many of the genes involved there are many cancers described. Maybe better to describe the cell physiological pathways in which they are involved in this column.

- HBV behaviour is different in Western and Asian cohorts and this may influence results (e.g.  Mueller-Breckenridge AJ, Garcia-Alcalde F, Wildum S, Smits SL, de Man RA, van Campenhout MJH, Brouwer WP, Niu J, Young JAT, Najera I, Zhu L, Wu D, Racek T, Hundie GB, Lin Y, Boucher CA, van de Vijver D, Haagmans BL. Machine-learning based patient classification using Hepatitis B virus full-length genome quasispecies from Asian and European cohorts. Sci Rep. 2019 Dec 11;9(1):18892. doi: 10.1038/s41598-019-55445-8.), This may require some discussion.

Author Response

Point 1. I think impact of this manuscript could be enhanced by referring in the introduction to recent efforts focused on surveilling patients with HBV for HCC development (e.g. Anugwom CM, Allaire M, Akbar SMF, Sultan A, Bollipo S, Mattos AZ, Debes JD.  Hepatitis B-related hepatocellular carcinoma: surveillance strategy directed by immune-epidemiology. Hepatoma Res. 2021;7:23. doi: 10.20517/2394-5079.2021.06.)

Response: We thank the reviewer for bringing this recent review to our attention. We have included HBV-HCC surveillance strategies used across various demographics in the introduction.

Point 2. Table 3 would benefit from listing the full length name of genes, also some emphasis on what is not found would be interesting, like integrations in Axin1/2/Tankyrase genes or so. Also the cancer associated annotations feel a little crude as for many of the genes involved there are many cancers described. Maybe better to describe the cell physiological pathways in which they are involved in this column.

Response: We have added the full-length name of each gene in Table 3 in accordance with the HNGC database. We agree that it would be interesting to assess what known HCC driver genes were not found by analyzing RTGs. However, this is not the focus of this study. Our approach can only identify driver genes involved in HBV-HCC (rather than HCC in general) and is limited to insertional mutagenesis. HBV Integration could contribute to carcinogenesis by causing chromosomal instability, expression of the oncogenic HBV protein X, and inflammation induced by other HBV proteins expressed from the integrated DNA. Although the most frequent RTG, TERT, is also the most frequently mutated gene in HCC in general, other frequently mutated HCC genes, such as TP53 and CTNNB1, were not among the top 28 RTGs. These findings suggest that using RTG analysis as a tool to identify driver genes is limited to HBV-HCC. This limitation is now mentioned in the revised manuscript (section 4). In Table 3, the purpose of annotation with known cancers a gene has been implicated in was to demonstrate that most RTGs have existing associations with carcinogenesis, rather than to list every such association. To provide better context in which to consider the potential HCC-related roles of the top RTGs (n=28), we have now added the results of pathway analyses of these genes with Enrichr GO Biological processes. We have identified 180 pathways as summarized separately in Supplemental Table S3 and mentioned in section 3.3.

Point 3.  HBV behaviour is different in Western and Asian cohorts and this may influence results (e.g.  Mueller-Breckenridge AJ, Garcia-Alcalde F, Wildum S, Smits SL, de Man RA, van Campenhout MJH, Brouwer WP, Niu J, Young JAT, Najera I, Zhu L, Wu D, Racek T, Hundie GB, Lin Y, Boucher CA, van de Vijver D, Haagmans BL. Machine-learning based patient classification using Hepatitis B virus full-length genome quasispecies from Asian and European cohorts. Sci Rep. 2019 Dec 11;9(1):18892. doi: 10.1038/s41598-019-55445-8.), This may require some discussion.

Response: We agree this is an important distinction. We have revised Table 1 to provide geographic context to the HBV cohorts included in the study. A majority of patient specimens were collected in Asian countries (72.7%, 24/33 studies), while only 6.0% (2/33) of the studies were conducted in Western populations. In seven studies, patient demographic characteristics were unavailable or specimens were procured commercially. We have included this information in the revised manuscript (section 3.2). 

Reviewer 3 Report

The manuscript by Lin et al. “Characterization of genes recurrently targeted by HBV integration in hepatocellular carcinoma: a potential tool for driver identification” summarizes genes that might be influenced by HBV integration in liver cancer. The basic hypothesis is that frequent HBV target sites are likely significant for HCC carcinogenesis thus can be driver genes for HCC. The fact that there are so many HBV target sites suggest that these integration sites are likely incidental, rather than causal for HBV development. The driver genes can be indeed regulated by HBV integration as evidenced by the example of TERT expression. However the significance of other genes can be further enhanced through correlation with their expression in HCC through analysis using TCGA datasets while the authors failed to do so. In addition, the current analysis does not rise above previous such efforts by answering questions such as number of integration sites in each HCC tumor as well as number of integration sites in individual tumor cells.

Author Response

The manuscript by Lin et al. “Characterization of genes recurrently targeted by HBV integration in hepatocellular carcinoma: a potential tool for driver identification” summarizes genes that might be influenced by HBV integration in liver cancer. The basic hypothesis is that frequent HBV target sites are likely significant for HCC carcinogenesis thus can be driver genes for HCC. The fact that there are so many HBV target sites suggest that these integration sites are likely incidental, rather than causal for HBV development. The driver genes can be indeed regulated by HBV integration as evidenced by the example of TERT expression.

Point 1. However the significance of other genes can be further enhanced through correlation with their expression in HCC through analysis using TCGA datasets while the authors failed to do so. In addition, the current analysis does not rise above previous such efforts by answering questions such as number of integration sites in each HCC tumor as well as number of integration sites in individual tumor cells.

Response: We agree that correlating recurrent genes with their expression in HCC patients is important and appreciate the suggestion to look into TCGA data. Unfortunately, we found a majority of cases in the TCGA HCC cohort to be either HBV-negative or to have undetermined HBV infection status, precluding a rigorous analysis of RTG expression. We also agree that no new data were generated. Nevertheless, we believe that a systematic overview of the available evidence can still facilitate identification of HBV-HCC drivers, aid in drug development, and improve disease management. We discuss these points in the last paragraph of the Discussion.

Round 2

Reviewer 1 Report

Point 1. Characterization of genes recurrently targeted by HBV integration in hepatocellular carcinoma: a potential tool for driver identification, the reader became bored with the verbosity.

Response: We have changed the manuscript title to Recurrent HBV integration targets as potential drivers in hepatocellular carcinoma”.

This is a good title.

Point 2. The abstract should rewrite, could not list too many results in it.

Response: We have revised the abstract accordingly and marked the changes in red.

Revised version  is better than previous one.

Point 3 Figure 1 and figure 2, figure 3 need regenerate. Should be clear and publishable.

Response: Figures 1-3 have been regenerated at 600 DPI.

Figure 1 is OK, figure 2 and 3  if would increase would be better.

Reviewer 3 Report

The authors provided improvement for their manuscript so that the results presented becomes more specific and informative for HBV-associated HCC.